# Geometrical model of lobular structure and its importance for the liver perfusion analysis

**Eduard Rohan**[1]*, **Jana Camprová Turjanicová**[1], **Václav Liška**[2]

**1** Department of Mechanics, Faculty of Applied Sciences, NTIS – New Technologies for Information Society, University of West Bohemia, Pilsen, Czech Republic, **2** Biomedical Center, Faculty of Medicine, Charles University Pilsen, Pilsen, Czech Republic

* rohan@kme.zcu.cz

**Data Availability Statement:** A) Geometrical model of the liver tissue periodic cell available at: https://doi.org/10.6084/m9.figshare.17000119 B) The computations of the perfusion using the SfePy

## Abstract

A convenient geometrical description of the microvascular network is necessary for computationally efficient mathematical modelling of liver perfusion, metabolic and other physiological processes. The tissue models currently used are based on the generally accepted schematic structure of the parenchyma at the lobular level, assuming its perfect regular structure and geometrical symmetries. Hepatic lobule, portal lobule, or liver acinus are considered usually as autonomous functional units on which particular physiological problems are studied. We propose a new periodic unit—the liver representative periodic cell (LRPC) and establish its geometrical parametrization. The LRPC is constituted by two portal lobulae, such that it contains the liver acinus as a substructure. As a remarkable advantage over the classical phenomenological modelling approaches, the LRPC enables for multiscale modelling based on the periodic homogenization method. Derived macroscopic equations involve so called effective medium parameters, such as the tissue permeability, which reflect the LRPC geometry. In this way, mutual influences between the macroscopic phenomena, such as inhomogeneous perfusion, and the local processes relevant to the lobular (mesoscopic) level are respected. The LRPC based model is intended for its use within a complete hierarchical model of the whole liver. Using the Double-permeability Darcy model obtained by the homogenization, we illustrate the usefulness of the LRPC based modelling to describe the blood perfusion in the parenchyma.

## Introduction

The liver as a vital organ plays in the human body a fundamental role in its numerous functions. Thus, any disease or pathological state of the liver tissue can cause serious health problems, even death of the patient and needs to be appropriately treated. The understanding of liver perfusion on the multiple scales is crucial for the surgical treatment as the liver resection, transplantation but also for understanding how the liver perfusion is modified by diffuse parenchymatous diseases such as cirrhosis, steatohepatitis, sinusoidal obstruction syndrome (last two are connected with modern chemotherapies regiment), etc [1, 2]. The need of understanding of liver perfusion and also liver regeneration, which is dependent upon perfusion status,

software – the script files available at: Skript: https://doi.org/10.6084/m9.figshare.17000881.

**Funding:** This research was supported by the European Regional Development Fund-Project "Application of Modern Technologies in Medicine and Industry", No. CZ.02.1.01/0.0/ 0.0/17 048/ 0007280. The funders had no role in study design, data collection and analysis, decision to publish, or preparation of the manuscript.

**Competing interests:** The authors have declared that no competing interests exist.

comes from the progress of liver surgery. Now we are able to perform extended liver resection with large loss of future liver remnant volume, stage procedures combined with portal vein embolization and also repeated procedures for recurrence of malignant liver lesions [3–5]. All these procedures are absolutely dependent upon radiological paging and remnant volume computation (i.e. CT volumetry) [6]. In case we are able to predict the liver regeneration capacity and also future achieved volume of remnant liver parenchyma, we could perform named surgical procedures also at the patients with diseases modifying liver regeneration.

At the macroscopic scale the liver receives blood from the two separated vascular system, one belonging to the hepatic artery and the other to the portal vein. These two vascular trees branch repeatedly, until they reach the microcirculation at the level of so-called hepatic units, typically considered as hexagonal lobules mutually separated by thin vascular septum, [7]. At this level, the blood from both the supplying vascular trees is mixed in the small hepatic capillaries also called sinusoids which enable for the the metabolic activity of hepatocytes surrounding these microvessels with fenestrated walls. From the sinusoids, the blood is drained to the central veins, which constitute another vascular network drained in the hepatic veins. Computational modelling of flows in this system of precapillary vessels and fenestrated capillaries was treated in a number of works using the Computational Fluid Dynamics (CFD) tools with non-Newtonian fluid rheology and constructed idealized geometrical models [8], or image-based models [9]. Motivated by understanding the physiological background of the sinusoidal network organization, in works [10, 11] a constructive algorithm was proposed to generate the hepatic capillaries within the whole lobule volume based on modelling the transport and metabolism processes.

*In silico* models can be a powerful tool to study liver perfusion on multiple scales [12, 13]. Existing models of blood flow in liver are more focused on the macrocirculation in the trees of hepatic artery and portal vein; besides the image-based approaches [14], various strategies have been proposed to construct branching structures, also called dendritic architectures [15], including about 20 bifurcation generations. While the global constrained constructive optimization [16], cf. [17] is based on purely geometrical features, algorithms incorporating more or less physiology-based rules [18] can account also for evolutionary processes. Recently, a few computational studies were made on the microcirculation determining perfusion between the portal track and the central vein at lobular level. These studies usually consider the conceptual hexagonal liver lobule as the hepatic functional unit, that reflects only anatomical conditions. For computer simulations of normal and pathological liver perfusion, we have to accept physiological view of this problem [19]. Papers published until now were working only with healthy liver, however, the future and most important impact of computational modelling is in diagnostics of diffused liver diseases, such as liver cirrhosis, steatofibrosis, where the fibrotisation of the tissue (septic parts mostly) is crucial for understanding the problem and for helping solve clinically important questions. In this respect, irregularities in blood flows and hepatic functionality in neighboring lobules should be taken into account [20, 21].

Using the poroelastic theory [22], developed an idealized 2D model of the blood flow in the longitudinal section of lobules. Siggers *et al.* [23] modelled the blood interstitial flow in the liver tissue, represented by a lattice of hexagonal lobules. The lobules were assumed to be long enough to neglect the effect of their ends and thus only a 2D problem was considered. The 2D model of microcirculation within a single liver lobule, which admits a blood inflow via the vascular septa connecting adjacent portal segments, was presented by [24] in context of cytokine distribution. Höhme *et al.* [25] presented the idealized model of lobule with the basic unit to be individual hepatocyte in the context of liver regeneration and restoration of micro-architecture. This model was extended in recent work [26], presenting more complex 3D simulations of disease pathogenesis through liver tissue. However, the modeling was still based on the

classical concept of one hexagonal lobule. Deabbaut *et al.* [7] focused on the perfusion modeling in the sinusoids based on the micro-CT images of real 3D sinusoidal geometries of human liver, using CFD methods. This model was recently used in multilevel modeling of hepatic perfusion [27] as a tool to obtain perfusion characteristic for modeling of single lobule, whereby importance of the interlobular septa in modeling of the lobule perfusion has been discussed. Only rarely, in the papers mentioned above, the importance of the interlobular septa for the interlobular permeability has been discussed and aspects of increased septic fibrotisation that remarkably influences this permeability was neglected. Zonal variation of the sinusoidal network and the related microdosimetry was studied in [19] using the single hepatic lobule, in the context of the microcirculation related to the hepatocyte functionality.

## Aims

The modelling of microcirculation in the liver is based usually on the classic hexagon representation of lobule. Although these structures are presented in the liver parenchyma, their role as the autonomous hepatic functional units is at least questionable [5]. Therefore, we propose a two-scale approach to the liver perfusion modelling which respects the hexagonal lobular structure, however, drops any assumption of the autonomous functional units.

In this paper, we pursue the following aims:

- To propose a concept of the geometric structure of the liver parenchyma at lobular level as an integral part of a complex multiscale model of the blood perfusion and related coupled physiological processes. In particular, relaxing the usual symmetry assumption on the hepatic functional unit should enable to assess the role of the interlobular septum permeability potentially influenced by diffusion diseases.

- To create a parametric model of the liver lobule which generates a periodic three dimensional (3D) structure. This will allow for studying the perfusion response sensitivity with respect to morphological parameters of the tissue microstructure.

- As a proof of concept, to apply one of the tissue perfusion models previously developed using the homogenization method [36] to the periodic liver structure generated by the LRPC. The aim is to demonstrate qualitative features of the two-scale computational analysis of the liver perfusion.

The LRPC based models of the liver parenchyma derived using homogenization techniques can serve only a part of hierarchical models of the whole liver. The other part to be connected with comprises convenient models of the blood flow at higher level vasculature associated with the portal and hepatic veins, and hepatic artery, cf. [14, 17, 28].

## Methods and models

### Functional units of the liver—Geometrical idealizations

The liver parenchyma at the mesoscopic level with characteristic lengths $\approx 100 \mu$m is constituted by functional units which are defined in terms of the principal microvascular vessels associated with portal vein (PV), hepatic vein (HV), and hepatic artery (HA). In an idealized geometrical model, PV and HV can be represented as straight parallel vessels (cylindrical tubes).

In a transversal plane, orthogonal to these vessels, these functional units are schematically depicted in Fig 1:

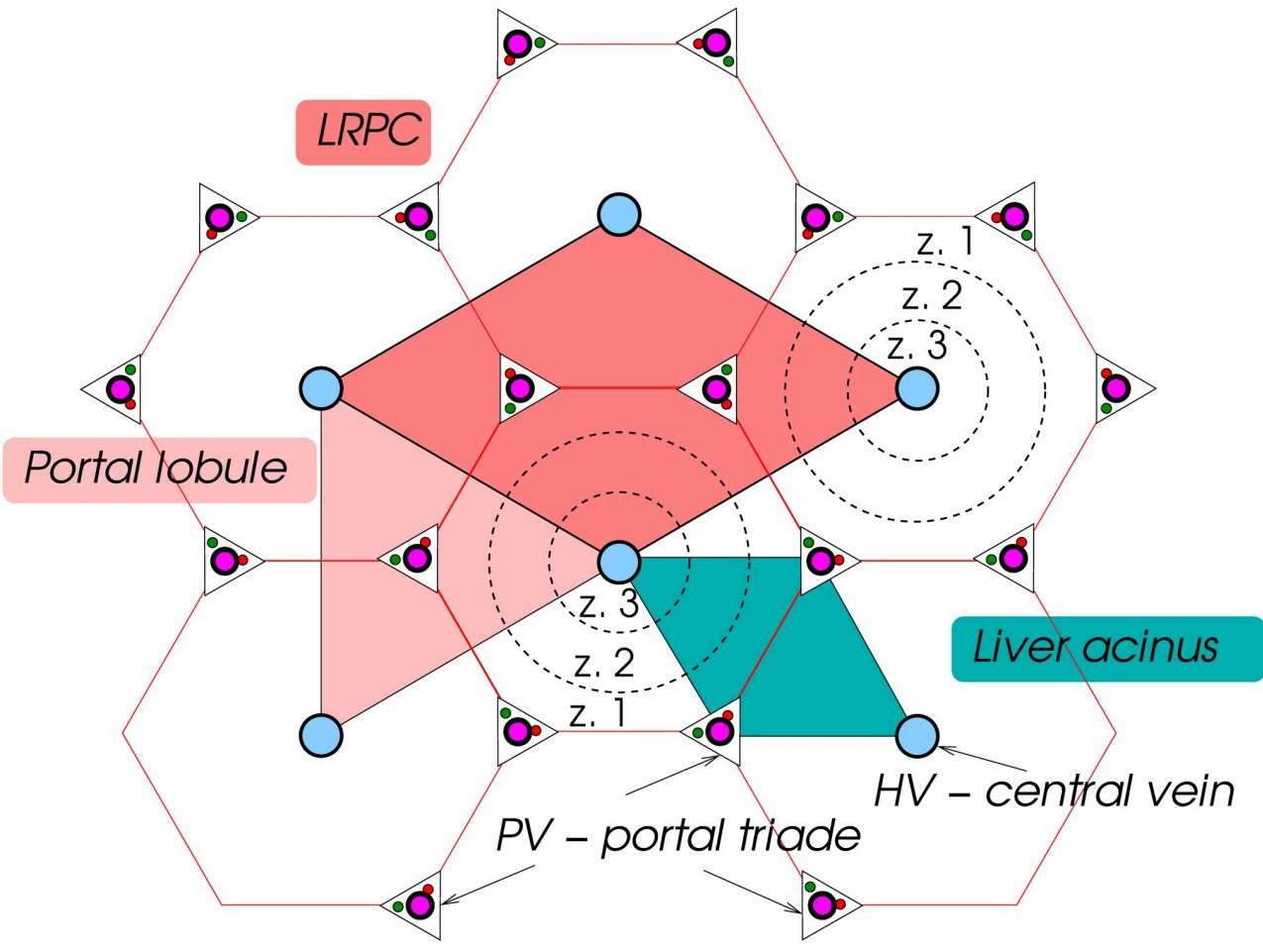

**Fig 1. Liver microstructure and functional units.** The liver representative periodic cell (LRPC) incorporates the Portal lobule and also the Acinus. Functional zones z. 1–3 are indicated.

- Hepatic lobule is the regular hexagon with its center determined by the terminal branch of HV (often called the central vein) and six vertices constituted by terminal branches of the PV, often called the vertex vessels, cf. [8]. Most often, geometrical model of the liver parenchyma is introduced as regular honeycomb structure generated by the hepatic lobule.

- Portal lobule is the regular triangle determined by any three HV, whereas the barrycenter is identified with the portal triad comprising PV, HA and the bile duct.

- Liver acinus is the smallest physiologically autonomous unit; it is a triangle whose vertices are defined by two closest PV (incorporated in portal triads) and by the central vessel (HV).

This 2D representation can be elevated to a 3D model where each lobule generates a hexagonal polyhedron. Interconnecting vessels between neighboring PV constitute the so-called vascular septum (VS) identified with faces of the above mentioned lobular polyhedron. The presence of VS influences flows from the PV to HV. In particular, VS is more permeable than the sinusoids, so that certain amount of the blood flows from the PV to VS and then from VS to HV. Effectively this phenomenon contributes to a better exploitation of the sinusoids

transport capacity, [29], and the metabolic processes. Within acinus, three metabolic zones are distinguished according to different rates of oxygen and nutrients consumption, usually denoted by numbers 1, 2, and 3, see [30], also indicated in Fig 1.

The above idealized picture of the liver parenchyma is a basis for introducing the so-called *liver representative periodic cell* (LRPC) which can generate the liver tissue as a periodic lattice. In reality, the vasculature at the lobular level is the terminal part joining the perfusion trees associated with the PV, HV and HA, so that the liver parenchyma can only be approximated as a locally periodic structure.

In addition to the above described functional units, a conical microvascular subunit of classic hexagonal lobule has been proposed in the literature [31], being called the primary lobule; it consists of a group of sinusoids supplied by a single portal track and its associated branches, [32].

## Mathematical description of the periodic lobular structure

As the main contribution of the paper, we introduce the ground structure generating two precapillary periodic vasculatures and define the liver representative periodic cell (LRPC) which enables the flow modelling using the periodic homogenization method [33, 34]. To show, how the geometric model can be applied, below we present an illustrative example of the flow modelling using the so-called double-porosity media approach [35–37].

**Perfusion modelling using the homogenization approach.**   The existing mathematical models are relevant purely to the lobular level of the liver, so the simulations of flow and other transport processes are performed for domains constituted by one, or several hexagonal lobules [20], whereby artificial boundary conditions have to be specified. Our approach is different. We aim to treat the perfusion as the two-scale problem, so that the macroscopic and microscopic phenomena are coupled by virtue of the homogenization method [36, 38, 39]. The macroscopic pressures associated with the terminal PV and HV branches may vary with the macroscopic position at the organ level, even though the lobular structure is described as a periodic lattice generated by the LRPC. As an advantage, the so-called characteristic response of the LRPC enables to establish effective coefficients of the perfusion model, so that the computation relevant to the macroscopic scale reflects geometrical features of the microstructure.

The most obvious periodical structure in the liver tissue are hepatic lobules. However, as stated above, other functional units, namely the portal lobule and acinus are more important for description of the liver physiology. Therefore, for the mathematical construction of the periodic structure, we propose the smallest possible cell, which contains both these units and which generate a periodic layout quite straightforwardly. We consider a rhombus unit involving two portal tracks and four central veins, one situated at each corner, see Fig 2. This unit is formed by two neighboring portal lobules and contains one whole acinus. Using the geometrical parameters of the hepatic lobule taken from the literature, *e.g.* [23], we may construct the LRPC geometry and define a permeability of the sinusoidal porosity representing the capillary network [40].

**Geometry of LRPC and periodic lobular structure.**   The proposed model of the lobular structure is constituted by the principal and transversal cylindrical vessels. Both these groups of vessels are involved in the two distinguished vasculatures $\mathcal{Y}_p$ and $\mathcal{Y}_h$ connected to the portal vein (PV) and to the hepatic vein (HV), respectively. Below we introduce the primary and dual lattices which enable us to construct the two mutually disjoint vasculatures associated with the central and the vertex veins. The primary lattice is used to define the periodic lobular structure. It is characterized by the representative periodic cell occupying domain $Y$ which is decomposed into three disjoint parts $Y_h$, $Y_p$ and $Y_m$, associated with the precapillary parts of HV, PV,

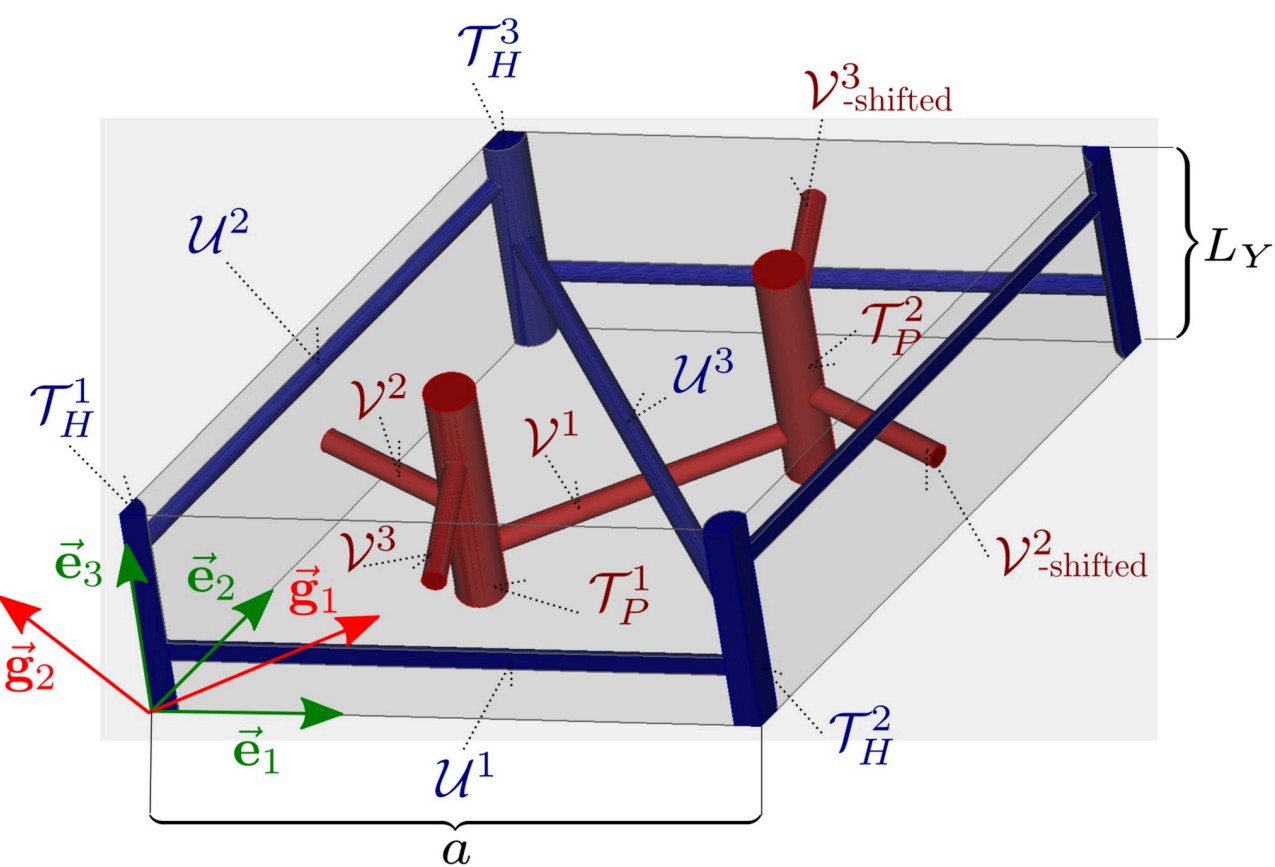

**Fig 2. The LRPC geometry with descriptions of the PV and HV tracks connected with the central and vertex veins of the lobule.**

and with the sinusoidal porosity (the matrix), respectively, such that $Y = Y_h \cup Y_p \cup Y_m$, see Figs 2 and 7.

In what follows, we employ the notation and geometrical entities introduced formally in the Appendix. The LRPC generating the periodic lattice is based on two ground structures. The first one is constituted by points $\{h^i\}_i$, $i = 1, \ldots, 4$, see Eq (16) and Fig 3, which define positions of the central *hepatic* veins. Analogously, points $\{p^i\}_i$, $i = 1, \ldots, 4$ constituting the second ground structure determine positions of the vertex *portal* veins, see Eq (17). The central vein centered at point $h^4$ is not involved in the portal lobulus, see Fig 4, but is needed to establish the LRPC and transverse vessels interconnecting the central vein tracks, see below. These two groups of the points are introduced using the *primary* and the *dual* lattices associated with two parameters $\bar{a}$ and $\bar{b} = \bar{a}\sqrt{3}/3$ and defined in terms of the bases $\{e^k\}_k$ and $\{g^k\}_k$, $k = 1, \ldots, 3$, respectively, see Eq (15).

**Precapillary vessels $Y_h$ and $Y_p$.** Structures $\mathcal{Y}_h$ and $\mathcal{Y}_p$ consist of the principal and transversal vessels constituted by cylinders $\mathcal{T}$ which are defined by a position $x$, a unit vector $n$ determining the vessel axis, and by the vessel length $L$ and radius $R$, see Fig 5,

$$\mathcal{T}(x, n, L, R) = \{y \in \mathbb{E} | 0 < d(y, x) \cdot n < L, \, |r(d)| < R\},$$

$$\text{where} \quad d(y, x) = y - x,$$

$$r(d) = d - (n \cdot d)n,$$

(1)

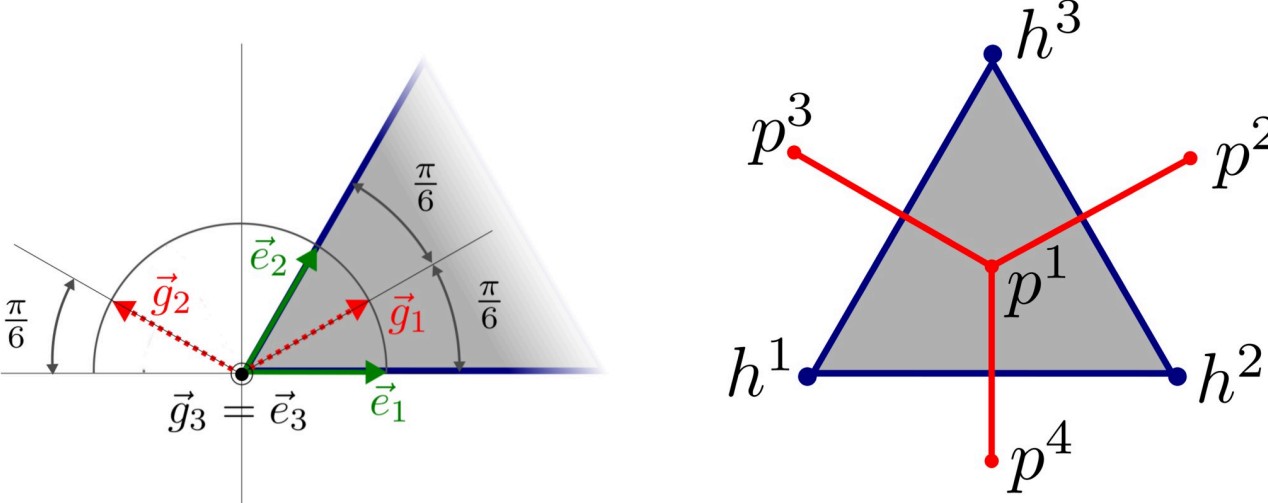

**Fig 3. Positions of the nodes generating the primary and dual lattices.** Left: The generating unit is defined using unit vectors of the primary and dual bases, $\{e^i\}$ and $\{g^i\}$, respectively. Right: position of the vertex $p^k$ and the central $h^k$ points within the generating unit—the portal lobule.

where the radial relative position $r$ with respect to the vessel axis is evaluated using the projection of $d$ into $n$.

The principal vessels include the hepatic (central) veins $\mathcal{T}_H$, and the portal (vertex) veins $\mathcal{T}_P$ which are generated according to Eq (1),

$$\mathcal{T}_H^k(h^k, e_3, L_H, R_H) \;, k = 1, \cdots, 4 \;, \quad \mathcal{T}_P^l(p^l, e_3, L_P, R_P) \;, l = 1, \cdots, 4 \;. \tag{2}$$

Note that the portal lobule is constituted by 3 veins $\mathcal{T}_H^k$, with $k = 1, 2, 3$ and one $\mathcal{T}_P^1$, only, however, four vessels of the two groups are employed below to define the LRPC.

The length of both the vessel types have equal lengths $L_H = L_P = \bar{L}$ equal to the thickness of the representative cell $Y$. The central vein vessels are interconnected by transversal cylindrical vessels $\mathcal{U}^i$, $i = 1, \ldots, \bar{i}$. Each of them is determined by positions $y_i^A$ and $y_i^B$ of the end nodes situated on axes of two central veins, see Eq (3), being parameterized by $\xi_i^A$, $\xi_i^B \in [0, \bar{L}]$ and by the radius $R_i$. For the $i$-th vessel, Table 1 provides the two links $(A, i) \mapsto k$ and $(B, i) \mapsto l$ so that, the specific couple $(A, B)$ is associated with two particular central hepatic veins labeled by

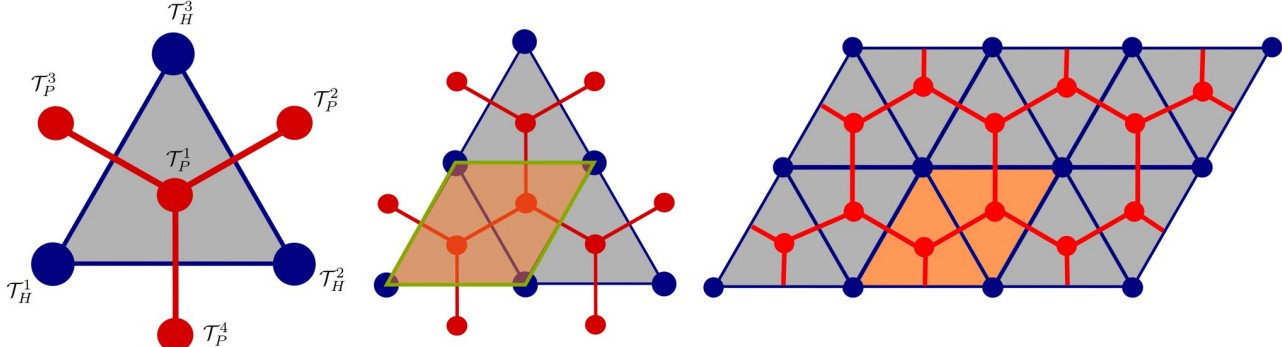

**Fig 4. Construction of the LRPC.** Left: Portal lobule unit associated with $\mathcal{Y}_d$, $d = h, p$. Middle & Right: The LRPC is obtained by translations, see Eqs (5) and (6), and generate the periodic lattice.

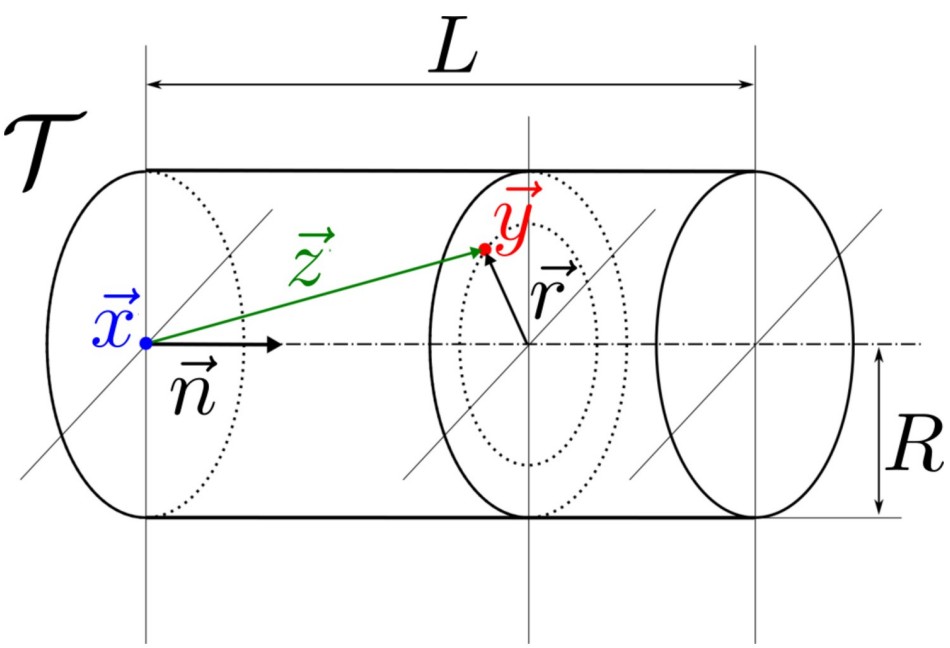

**Fig 5. Generating vessel $\mathcal{T}(\boldsymbol{x}, \boldsymbol{n}, L, R)$, see Eq (1).**

$k, l \in \{1, \ldots, 4\}$. The same expression Eq (1) is employed to define $\mathcal{U}^i(\boldsymbol{y}_i^A, \boldsymbol{n}_i, L_i, R_i)$, whereby

$$\boldsymbol{y}_i^A = \boldsymbol{h}^k + \xi_i^A \boldsymbol{e}_3 \ , \ \ \boldsymbol{y}_i^B = \boldsymbol{h}^l + \xi_i^B \boldsymbol{e}_3 \ , \ \ L_i = |\boldsymbol{y}_i^B - \boldsymbol{y}_i^A| \ , \ \ \boldsymbol{n}_i = (\boldsymbol{y}_i^B - \boldsymbol{y}_i^A)/L_i \ . \tag{3}$$

In analogy, the vertex veins are interconnected by transversal cylindrical vessels $\mathcal{V}^j$, $j = 1, \ldots, \bar{j}$, see Table 2. The $j$-th vessel of this group is parameterized by $\eta_j^A, \eta_j^B \in 0, \bar{L}$, associated with two particular indices of vertex portal veins, and by the radius $R_j$. Table 2 provides the two links $(A, j) \mapsto k$ and $(B, j) \mapsto l$. Vessel $\mathcal{V}^j = \mathcal{T}(\boldsymbol{y}_j^A, \boldsymbol{n}_j, L_j, R_j)$ is defined according to Eq

**Table 1. Transversal vessels $\mathcal{U}^i$, $i = 1, 2, 3$ of the central vein vasculature.** The structure is generated by three vessels, see Eq (3).

| #/$i$ | $A/k$ | $B/l$ | $\xi^A$ | $\xi^B$ | R |
|---|---|---|---|---|---|
| 1 | 1 | 2 | $0.3\bar{L}$ | $0.3\bar{L}$ | $0.1\bar{a}$ |
| 2 | 1 | 3 | $0.8\bar{L}$ | $0.8\bar{L}$ | $0.1\bar{a}$ |
| 3 | 2 | 3 | $0.5\bar{L}$ | $0.5\bar{L}$ | $0.1\bar{a}$ |

**Table 2. Transversal vessels $\mathcal{V}^j$, $j = 1, 2, 3$ of the vertex vein vasculature.** The structure is generated by three vessels, see Eq (4).

| #/$j$ | $A/k$ | $B/l$ | $\eta^A$ | $\eta^B$ | R |
|---|---|---|---|---|---|
| 1 | 1 | 2 | $0.1\bar{L}$ | $0.1\bar{L}$ | $0.12\bar{b}$ |
| 2 | 1 | 3 | $0.4\bar{L}$ | $0.4\bar{L}$ | $0.12\bar{b}$ |
| 3 | 1 | 4 | $0.7\bar{L}$ | $0.7\bar{L}$ | $0.12\bar{b}$ |

(1), where

$$\boldsymbol{y}_j^A = \boldsymbol{p}^k + \eta_j^A \boldsymbol{e}_3 \ , \ \ \boldsymbol{y}_j^B = \boldsymbol{p}^l + \eta_j^B \boldsymbol{e}_3 \ , \ \ L_j = |\boldsymbol{y}_j^B - \boldsymbol{y}_j^A| \ , \ \ \boldsymbol{n}_j = (\boldsymbol{y}_j^B - \boldsymbol{y}_j^A)/L_j \ . \tag{4}$$

Using the principal vessels $\mathcal{T}_H^k$, $\mathcal{T}_P^l$ and the transverse vessels $\mathcal{U}^i$, $\mathcal{V}^j$ established above in Eqs (2)–(4), we can construct the precapillary vascular networks of the portal and hepatic veins,

$$\begin{aligned} \mathcal{Y}_h &= \bigcup_{k=1,2,3} \mathcal{T}_H^k \cup \mathcal{U}_H \ , \\[4pt] \mathcal{U}_H &= \bigcup_{i=1,2,3} \mathcal{U}^i \cup \hat{\mathcal{U}}^1 \cup \hat{\mathcal{U}}^2 \ , \quad \text{where } \hat{\mathcal{U}}^j = \mathcal{U}^i + \bar{a}\boldsymbol{e}_j \ , \ j = 1, 2 \ , \end{aligned} \tag{5}$$

so involving translated vessels $\hat{\mathcal{U}}^j$, see Fig 6, and

$$\begin{aligned} \mathcal{Y}_p &= \bigcup_{l=1,2,3} \mathcal{T}_P^l \cup \mathcal{V}_P \ , \\[4pt] \mathcal{V}_P &= \bigcup_{j=1,2,3} \mathcal{V}^j \cup \hat{\mathcal{V}}^2 \cup \hat{\mathcal{V}}^3 \ , \quad \text{where } \hat{\mathcal{V}}^2 = \mathcal{V}^2 + \bar{a}\boldsymbol{e}_1 \ , \ \hat{\mathcal{V}}^3 = \mathcal{V}^3 + \bar{b}(2\boldsymbol{g}_1 + \boldsymbol{g}_2) \ . \end{aligned} \tag{6}$$

**Periodic lattice based on the LRPC**. The LRPC occupying domain $Y$ is defined within the primary lattice based on the ground structure of the central vessels,

$$Y = \{\boldsymbol{y} \in \mathbb{E} \mid \boldsymbol{y} = \sum_{i=1}^3 c^i \boldsymbol{e}_i \ , \ 0 < c^k < \bar{a}, \ k = 1, 2 \ , \ 0 < c^3 < \bar{L}\} \ . \tag{7}$$

It can be seen, that $Y$ corresponds anatomically to two adjacent portal lobules and, thus, contain one whole acinus. Indeed, in the plane determined by vectors $\boldsymbol{e}_1$ and $\boldsymbol{e}_2$, the liver acinus is the convex hull of the ground structure nodes $\boldsymbol{p}^1, \boldsymbol{p}^2, \boldsymbol{h}^2$ and $\boldsymbol{h}^3$.

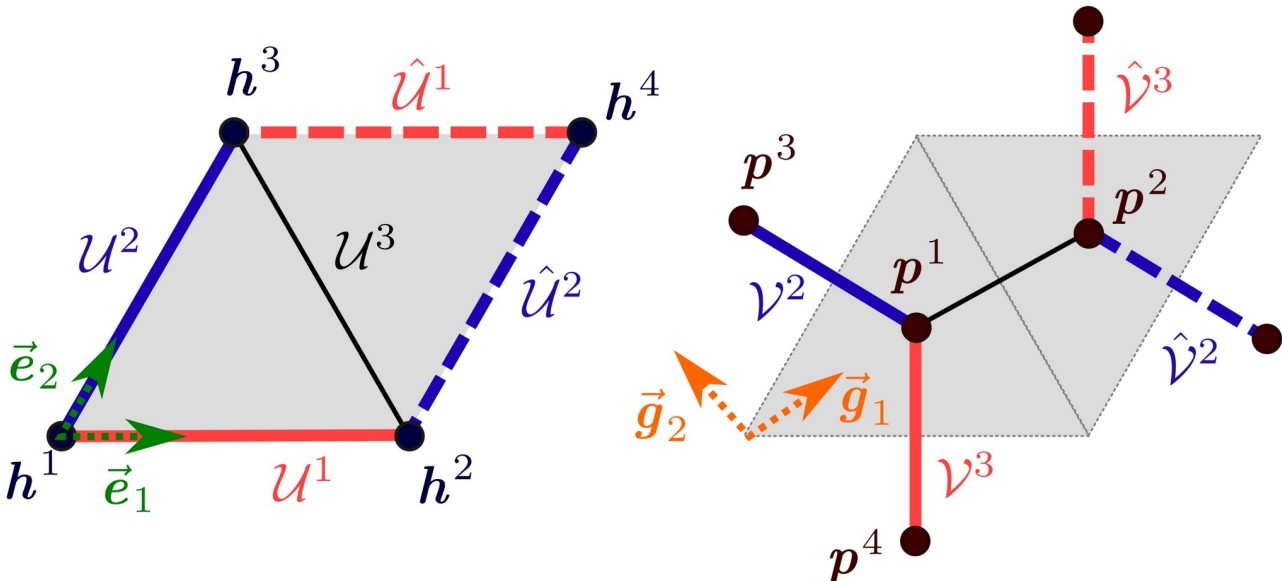

**Fig 6. Translations employed to define $\mathcal{Y}_p$ and $\mathcal{Y}_h$, see Eqs (5) and (6).**

The LRPC introduced in Eq (7) is decomposed in three disjoint parts $Y_h$, $Y_p$, and $Y_m$ which are defined, as follows,

$$
\begin{aligned}
Y_h &= \mathcal{Y}_h \cap Y \;, \\
Y_p &= \mathcal{Y}_p \cap Y \;, \\
Y_m &= Y \backslash (\overline{Y_h \cup Y_p}) \;.
\end{aligned}
\tag{8}
$$

The periodic structure of the liver tissue is generated by copies of the LRPC rescaled by parameter $\varepsilon > 0$ which determines the ration between the characteristic sizes of lobular structure ($\approx 10^{-4}$m) and the whole liver ($\approx 10^{-1}$m). While $\bar{a}$ and $\bar{L}$ are fixed and close to 1, hence the volume $|Y| = \bar{a}\bar{L}\sqrt{3}/2 \approx 1$, a real size of the hepatic lobule represented by domain $\varepsilon Y$ is given by $a^\varepsilon = \varepsilon\bar{a}$ and $L^\varepsilon = \varepsilon\bar{L}$.

For the homogenization-based modelling of the heterogeneous liver tissue we need periodic lattices, $\mathcal{L}_d^\varepsilon$, associated with the three subparts $d = m, h, p$. These are defined by translating $Y_d$ in the directions of the primary lattice,

$$
\mathcal{L}_d^\varepsilon = \varepsilon\mathcal{L}_d = \bigcup_{\boldsymbol{k}\in\mathbb{Z}^3}\varepsilon\left(Y_d + \sum_{i=1}^{3}k_i\boldsymbol{e}^i\right),
\tag{9}
$$

where the triplet $\boldsymbol{k} = (k_1, k_2, k_3)$ determines the placement of copies of $\varepsilon Y_d$. For a convenient "micromodel" describing blood flows in these lattices, a macroscopic model describing effective flows in the homogenized lobular structure can be derived by the limit analysis $\varepsilon \to 0$. An example is reported below.

## LRPC based model of flows in homogenized parenchyma

The blood perfusion in a liver tissue sample occupying a bounded domain $\Omega \subset \mathbb{R}^3$ can be described by an effective flow model obtained by the homogenization method. To this aim, at the heterogeneity level of the tissue decomposed in the three compartments represented by domains $\Omega_d^\varepsilon = \Omega \cap \mathcal{L}_d^\varepsilon$, $d = m, h, p$, see Eq (9), the so-called Double-permeability Darcy (DD) flow model can be employed. It is characterized by a special $\varepsilon$-scaling of the permeability associated with the dual porosity representing the sinusoidal network situated in $\mathcal{L}_m^\varepsilon$. In this section, we report equations of the effective (macroscopic) flow model which has been derived by the limit analysis $\varepsilon \to 0$ of the DD model [28], cf. [41]. Because of the double permeability effect associated with the sinusoidal, the limit model of the homogenized flow must be interpreted for a given $\varepsilon_0 \approx 10^{-3}$.

**Macroscopic equations.**  At the macroscopic scale, the flow in the lobular structure occupying a domain $\Omega$ is described in terms of two pressure fields, $p_h$ and $p_p$, associated with the portal and hepatic compartments. The parallel flows and the fluid exchange between these compartments is governed by the macroscopic equations

$$
\begin{aligned}
-\nabla \cdot \boldsymbol{K}_p\nabla p_p + G(p_p - p_h) &= f_p \;, \\
-\nabla \cdot \boldsymbol{K}_h\nabla p_h - G(p_p - p_h) &= f_h \;,
\end{aligned}
\tag{10}
$$

involving the so-called permeability tensor $\boldsymbol{K}_d = (K_d^{ij})$, $i, j = 1, \ldots, 3$, associated with the portal and hepatic compartments, labelled by $d = h, p$, and the so-called perfusion coefficient $G$ representing the permeability of the sinusoidal porosity. The r.h.s. terms $f_p$ and $f_h$ represent sources and sinks, respectively, connecting the parenchyma model with upper hierarchies of the PV

and HV vascular trees. Typically, $f_p$ and $f_h$ attain forms of distributions supported at locations of terminal branches of those trees, see the Discussion below and references [17, 28], cf. [14].

Both $\boldsymbol{K}_d$ and $G$ can be obtained using the periodic homogenization of the Darcy flow in the double porosity medium with the periodic structure generated by the LRPC $Y$. The sinusoids distributed in $Y_m$ form a microporosity characterized by a highly anisotropic permeability $\boldsymbol{D}_m$ which reflects the blood flow in the capillary network, cf. [7, 42]. Also in the precapillary vessels represented by channels $Y_h$ and $Y_p$, the flow is described approximately by the diffusion equations with the permeabilities $\boldsymbol{D}_h$ and $\boldsymbol{D}_p$ which are established as an approximation of the Poiseuille-Stokes flow.

To obtain a unique solution of Eq (10), boundary conditions (BCs) must be prescribed on $\partial\Omega$, the boundary of $\Omega$ (by $\partial D$ we denote the boundary of any open bounded domain $D \subset \mathbb{R}^3$). The most natural BCs describe imposed pressures $\bar{p}_p$ and $\bar{p}_h$ on two boundary segments $\Gamma_p \subset \partial\Omega$ and $\Gamma_h \subset \partial\Omega$ which may not be disjoint, in general. The rest of the boundary is considered as impermeable, so that

$$\begin{aligned} p_d &= \bar{p}_d \quad \text{on } \Gamma_d, \ d = p, h, \\ \boldsymbol{n} \cdot \boldsymbol{K}_d \nabla p_d &= 0 \quad \text{on } \partial\Omega \backslash \Gamma_d. \end{aligned} \tag{11}$$

If $\Omega$ represents the whole organ, $\Gamma_d$ may vanishes, so that solvability conditions impose constraint $\int_\Omega (f_p + f_h) = 0$ expressing the mass conservation.

**Effective parameters computed using the LRPC.** We explain, how the parameters $\boldsymbol{K}_d$ and $G$ are computed for a specific geometry of the periodic liver tissue generated by the LRPC. The boundary $\partial Y_d = \partial_\# Y_d \cup \Gamma_d$ consists of the "periodic" part, $\partial_\# Y_d = \partial Y_d \cap \partial Y$, and of the interface part, $\Gamma_d = \partial \mathcal{Y}_d \backslash \partial Y$, for $d = h, p, m$. Further, the notion of the so-called Y-periodicity is employed: any function $\psi(\boldsymbol{y})$ defined for $\boldsymbol{y} \in \mathbb{E}$ is Y-periodic, if $\psi(\boldsymbol{y} + \bar{a}\boldsymbol{e}_k) = \psi(\boldsymbol{y})$ for $k = 1$, 2, and $\psi(\boldsymbol{y} + \bar{L}\boldsymbol{e}_3) = \psi(\boldsymbol{y})$ for any $\boldsymbol{y}$.

With this notation in hand, we present two kinds of the boundary value problems for the characteristic responses $\varphi_d^i$, 1 = 1, 2, 3, and $\hat{\varphi}_m$. The first characteristic responses $\varphi_d^i$ describes the dependence of the Darcy flow in the portal ($d = p$), or the hepatic ($d = h$) vessels on the macroscopic pressure gradients $\nabla p_d$. Functions $\varphi_d^i$, $i = 1, 2, 3$ defined in $Y_d$, are Y-periodic and satisfy:

$$\begin{aligned} \nabla \cdot \boldsymbol{w}_d^i &= 0, \quad \text{in } Y_d, \ d = h, p, \quad \text{where } \boldsymbol{w}_d^i = -\boldsymbol{D}_d \nabla(\varphi_d^i + y^i), \\ v \cdot \boldsymbol{w}_d^i &= 0 \quad \text{on } \Gamma_d, \end{aligned} \tag{12}$$

where $\boldsymbol{v}$ is the unit normal vector outward to $Y_d$ and $\boldsymbol{D}_d$ is the permeability established for the channel systems $\mathcal{T}_p$ and $\mathcal{T}_h$. The second characteristic response $\hat{\varphi}_m$ describes the dependence of the Darcy flow in the sinusoidal porosity on the pressure difference between the vertex and central vein networks. Function $\hat{\varphi}_m$ is Y-periodic and satisfies:

$$\begin{aligned} \nabla \cdot \hat{\boldsymbol{w}}_m &= 0, \quad \text{where } \hat{\boldsymbol{w}}_m = -\bar{\boldsymbol{D}}_m \nabla \hat{\varphi}_m \quad \text{in } Y_m, \\ \hat{\varphi}_m &= \begin{cases} 0 & \text{on } \Gamma_h, \\ 1 & \text{on } \Gamma_p. \end{cases} \end{aligned} \tag{13}$$

Then, using the solutions of Eqs (12) and (13), the macroscopic flow coefficients are computed by the following two integrals,

$$
\begin{aligned}
K_d^{ij} &= \frac{1}{|Y|} \int_{Y_d} [\boldsymbol{D}_d \nabla(\varphi_d^i + z^i)] \cdot \nabla(\varphi_d^j + z^j) = 0 \ , \ \ d = h, p \ , \\
G &= \frac{1}{|Y|} \int_{\Gamma_h} v \cdot (\boldsymbol{D}_m \nabla \hat{\varphi}) \ ,
\end{aligned}
$$

(14)

where $|Y|$ is the volume of $Y$.

**Two-scale simulations of the liver perfusion.**  To illustrate, how the geometrical model can be used for the perfusion modelling, we report an example of the flow simulation in a domain representing a tissue specimen constituted by the periodic lobular structure. The finite-element mesh of the LRPC used to compute numerically the characteristic responses defined in Eqs (12) and (13) was obtained using the software GMSH [43] which enables to implement the geometry parametrization, see Fig 7.

The sinusoidal capillary network distributed in $Y_m$ form a microporosity characterized by a highly anisotropic permeability $\boldsymbol{D}^m$ involved in problem Eq (13). We used the results of [7, 42], from where the permeability tensors can be reconstructed locally with respect to cylindrical coordinate systems established with its "z"-axes (basis vector $\boldsymbol{e}_3$, see Fig 2) aligned with the central veins $\mathcal{T}_H^k$, $k = 1, \ldots, 4$. This yields four tensors $\boldsymbol{D}_m^k(y)$ at any point $y \in Y_m$, consequently a unique tensor $\boldsymbol{D}_m(y)$ employed in Eq (13) is obtained by linear interpolation using the barrycentric coordinates over each of the two portal lobulae involved in the LRPC, as determined by points $\{\boldsymbol{h}^1, \boldsymbol{h}^2, \boldsymbol{h}^3\}$ and $\{\boldsymbol{h}^2, \boldsymbol{h}^4, \boldsymbol{h}^3\}$.

By virtue of the Darcy flow homogenization with the high-contrast permeability ansatz leading to the DD model Eq (10), the dual permeability involved in Eq (13) is scaled by $\varepsilon_0^{-2}$, thus, $\bar{\boldsymbol{D}}_m = \varepsilon_0^{-2} \boldsymbol{D}_m$ if $Y_m$ is the zoomed cell, such that $|Y_m| \approx \bar{a}^2 \bar{L}$. On contrary, if the real-sized cell $\varepsilon_0 Y_m$ is used instead of $Y_m$, one employs $\bar{\boldsymbol{D}}_m := \boldsymbol{D}_m$.

Also in the precapillary vessels channels $Y_p$ and $Y_h$, see Eqs (5), (6) and (8), representing the primary porosity, the flow is described approximately by the diffusion equations with the permeabilities $\boldsymbol{D}_p$ and $\boldsymbol{D}_h$ which are established as an approximation of the Poiseuille-Stokes flow, whereby no rescaling applies. Since the precapillary vessels defining the channels $Y_d$, $d = p, h$ of both the PV and HV venous systems are defined as cylindrical tubes, for any $i$-the vessel of the two systems $\mathcal{Y}_p$ and $\mathcal{Y}_h$, see Eq (5), the axial permeability can be established by $\bar{K}^{i,\alpha} = \pi R_i^2(z)/8\mu$ with an equivalent fluid viscosity $\mu$. Hence, in the $i$-th vessel $\mathcal{T}_i$ involved in $\mathcal{Y}_d$, $d =$

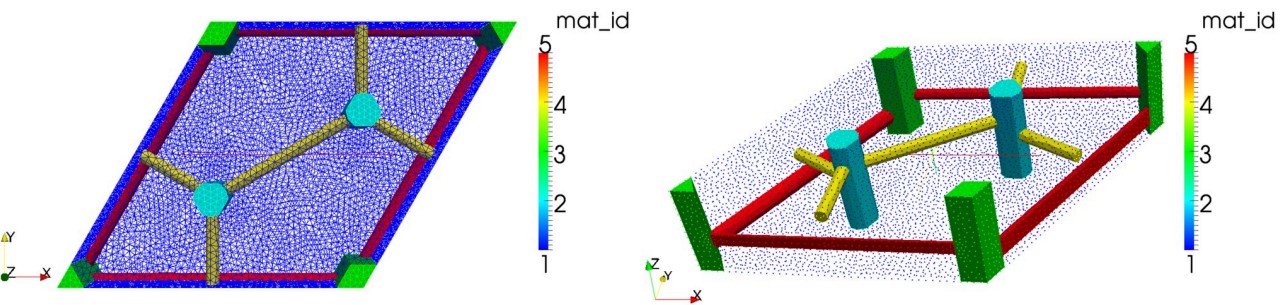

**Fig 7.**  Finite element mesh of the LRPC (right). Decomposition of domain $Y$ (left).

 

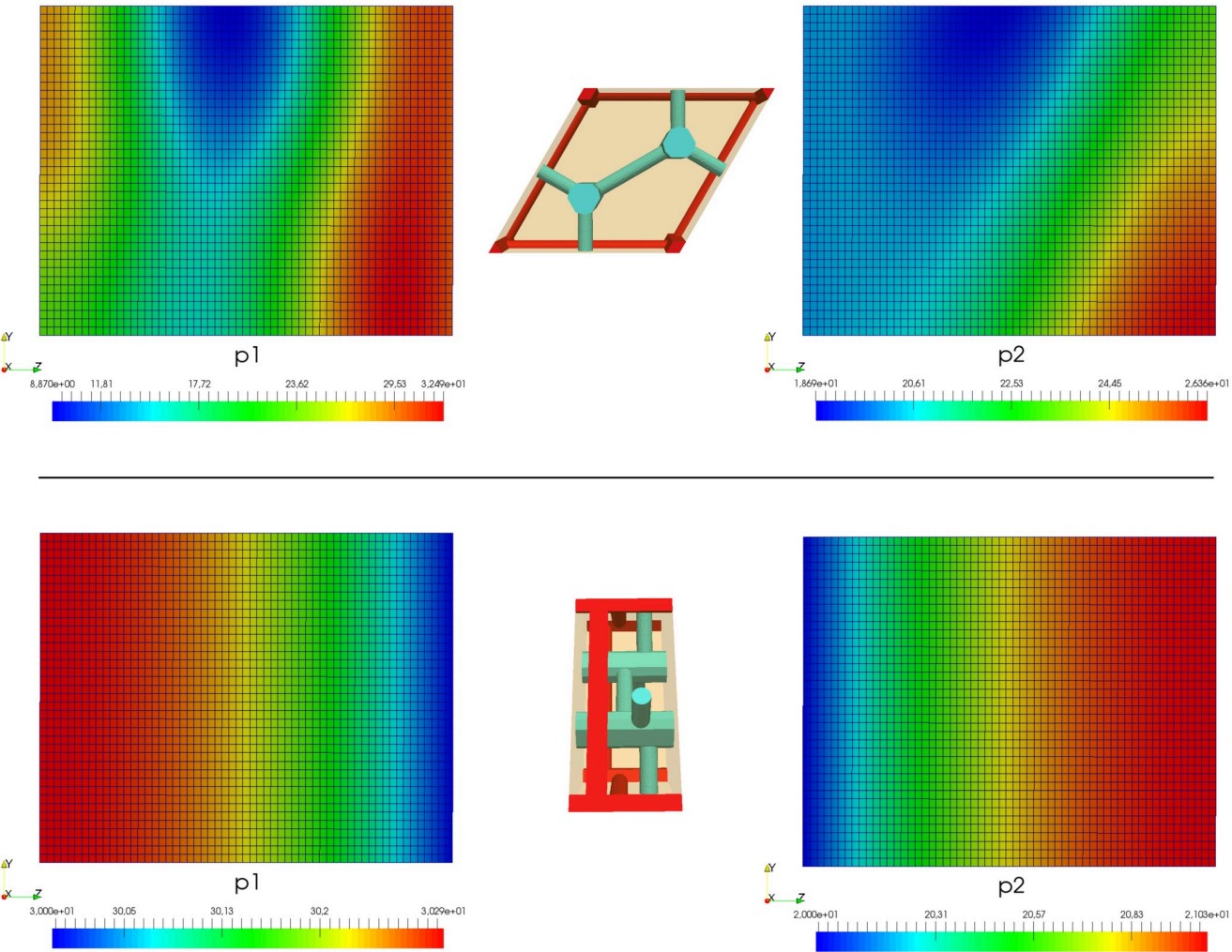

**Fig 8. Liver perfusion simulations for two different orientations of the lobular lattice (upper and lower rows).** The same boundary conditions according to Eq (11) are prescribed in both cases, whereby surfaces $\Gamma_p$ and $\Gamma_h$ form opposite sides of the specimen. Middle: orientations of the generating LRPC; Left: distributions of $p_p$; Right: distributions of $p_h$.

$p$, $h$, permeability $\boldsymbol{D}_d := \boldsymbol{D}^i$ involved in problem Eq (12) is defined by

$$\boldsymbol{D}^i(y) = \bar{K}^i \boldsymbol{v}^i \otimes \boldsymbol{v}^i + \kappa \boldsymbol{I}, \; y \in \mathcal{T}_i,$$

where $\boldsymbol{v}^i \otimes \boldsymbol{v}^i$ is the rank-one tensor generated by the vessel axial direction $\boldsymbol{v}^i$, while $\kappa \boldsymbol{I}$ is the isotropic permeability part given for a small regularization parameter $\kappa > 0$. Due to the tube overlaps in the vessel junctions, an average of involved permeabilities $\boldsymbol{D}^i$ computed for each relevant $i$ is taken. With permeabilities so established in channels $Y_d$, $d = p, h$, problems Eq (13) were solved.

In Fig 8, the macroscopic distributions of the pressures $p_p$ and $p_h$ are depicted for two different orientations of the lobular lattice. The same pressure slope on the opposite sites of the specimen was prescribed in both the cases, as well as other boundary conditions were considered identical. The different patterns are due to different orientations of the lobular structure. In Fig 9, the pressure and flow reconstructions are depicted for a selected macroscopic position.

 

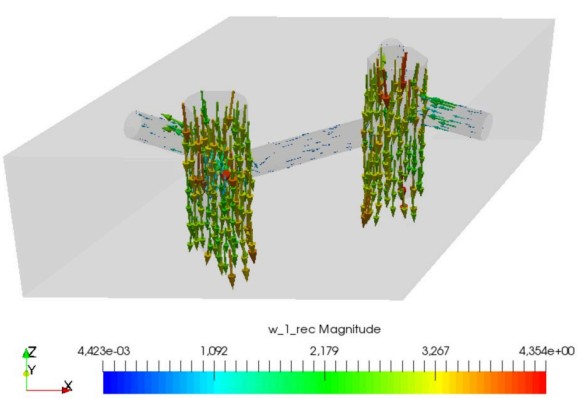

a) Portal vein channel system $Y_p$.

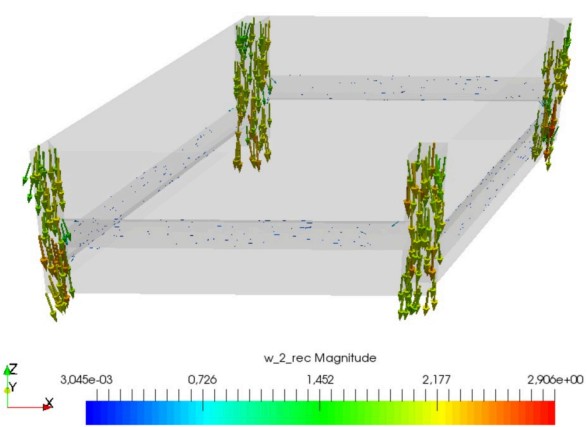

b) Hepatic vein channel system $Y_h$.

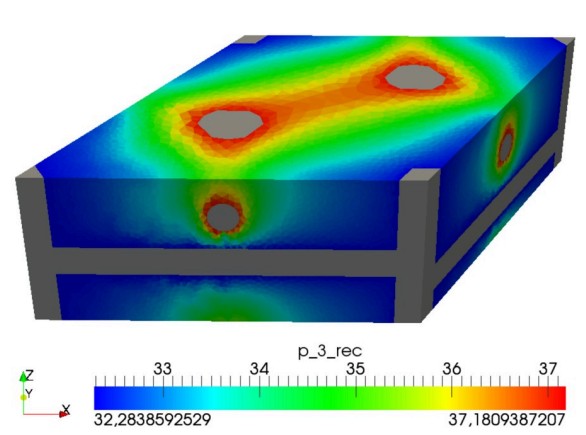

c) Pressure filed in sinusoids $Y_m$.

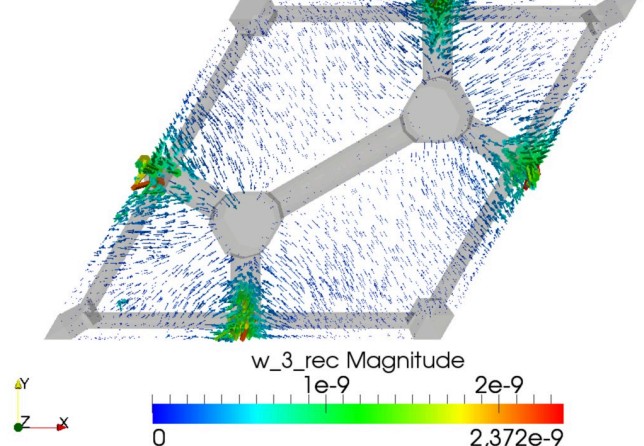

d) Velocity filed in sinusoids $Y_m$.

**Fig 9. Reconstructions of the microflows in a copy of the LRPC located at a selected point of the liver tissue specimen.** a),b) velocity fields in the precapillary vessels of the PV and HV systems. c),d) Sinusoidal flow.

## Discussion

The proposed geometric model of the periodic lobular structure is based on the LRPC which is associated with the primary lattice defined by the central hepatic veins. The LRPC is based on the union of two neighboring portal lobulae and contain one acinus, see Fig 1. The lobular structure representation based on the LRPC opens new perspectives of modelling physiological processes in the liver parenchyma. Let us recall that the postulated LRPC enables to generate a useful periodic structure of the liver tissue by the translation only, without necessity of any rotation.

The following advantages of the proposed model over the existing conceptions based on the standard functional units (the hepatic lobule, portal lobule, or the acinus) should be distinguished:

1. **Macroscopic gradients, inhomogeneity**. Up to our knowledge, the existing computational studies, cf. [23, 24, 44], are based on the domain of the hexagonal hepatic lobulus and assume a) all symmetries of the geometry, material parameters within this functional unit, and b) the macroscopic homogeneity. With these limitations, the lobular perfusion, oxygen and nutrient transport, and other physiological processes can be described only locally without relevancy to the macroscopic scale. Indeed, the symmetry assumption excludes in advance any influence of the macroscopic gradients of quantities of interest. Althoug the above mentioned computational studies bring a valuable insight into the lobule functionality, the restrictive assumptions disable to capture more realistic situations reflecting localized pathological or states after hepatectomy when hydraulic relations in the liver are disbalanced significantly. Obviously, any localized defect perturbs the whole system, such that the symmetry assumption allowing to reduce the computations one lobule cannot be accepted. On the contrary, the proposed LRPC based homogenization upscaling provides an avenue for multiscale modelling approaches which enable to respect the influence of nonuniform blood flows in the whole liver on the unevenly distributed pressure in different vertex veins of the lobules, thus, generating a nonsymmetric velocity field. By the consequence, this flow nonuniformity affects physiological processes related to the transport and concentrations of species dissolved in the blood. In particular, the change of the blood flow in liver after a surgical intervention (hepatectomy) can be described [12]. This is an important aspect to be respected namely when dealing with regeneration processes. Therefore, it is challenging to apply the LRPC based homogenization also in the context of earlier works [22, 45], where the phenomenological approach based on the Theory of Porous Media has been employed to simulate the perfusion and metabolic processes in one lobule only, or in a group of several lobules, adhering the symmetry assumptions.

2. **Irregular structure and generalization of the LRPC**. Due to its construction and the geometry parametrization, the LRPC design can capture large complexity of the precapillary vasculature. In particular, the number of the transversal vessels $\mathcal{U}^i$ and $\mathcal{V}^j$, as well as their sizes ans shapes can vary within one unit represented by $\mathcal{Y}_h$ and $\mathcal{Y}_p$, see Eq (5). This option enables to respect the vascular septum, cf. [27]. Moreover, it is possible to create aggregates by translation of the single LRPC in the principal lattice directions, so that larger periodic cells consisting of subunits are constructed. As shown in Fig 10, such aggregates can then be subject to a spatial transformation by a mapping which is periodic on the boundary of the created aggregates. Upon introducing further topology modifications to aggregated generating units $\mathcal{Y}_h$ and $\mathcal{Y}_p$ of the precapillary vessels, even more realistic microstructures can be created which capture more imperfections of the lobular structure. For instance, hexagons can be commuted by pentagons.

3. **Sinusoidal porosity**. Individual hepatic capillaries are not distinguished within the proposed LRPC geometric model, although such an extension is possible. Instead, the sinusoidal microporosity geometry can be respected in the context of the hierarchical homogenization by a locally periodic ultrastructure associated with a subscale, as proposed *e.g.* in [39], cf. [46]. As described above for the perfusion model, the hydraulic permeability can be introduced with respect to the radial, circumferential and axial directions, see e.g. [7] clearly defined through the abscissas $h_i$ of the hepatic lobule centered at the HV.

4. **Spatial grading of "quasi-periodic" and evolving microstructures**. The regular structure generated by a single LRPC can be subject to a geometrical transformation to create a graded structure. In such a case, the LRPC can still be used, since such a transformation is respected by the homogenization technique applied to any continuum model of flow, cf.

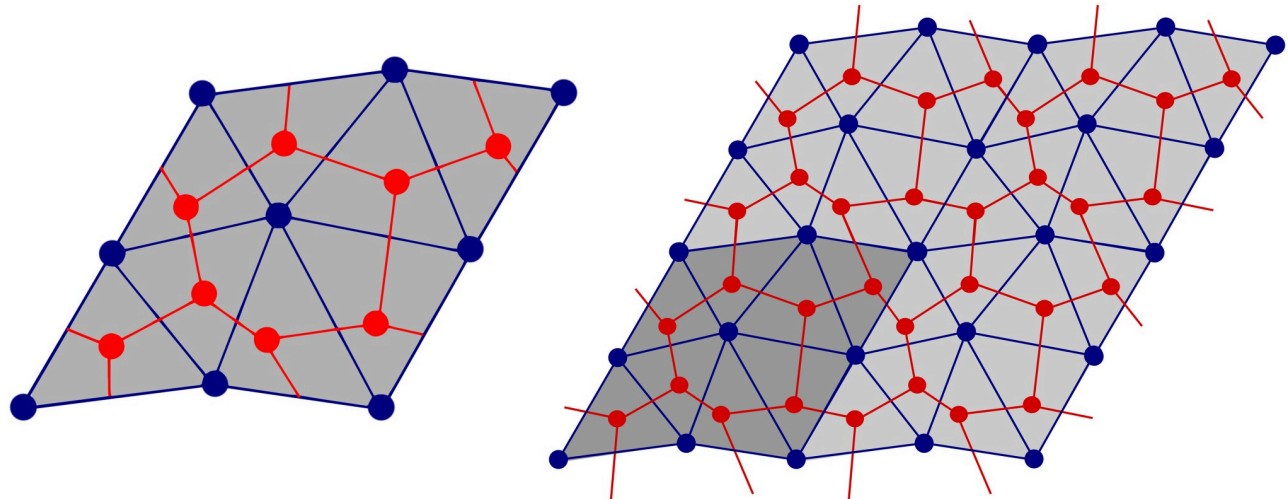

**Fig 10. Towards realistic lobular structures.** Left: The aggregate of 4 LRPC consisting of two portal lobules enables to create a larger periodic unit (the dark gray area), to consider nonuniform perturbations of the regular periodic structure which can comprise hexagonal and pentagonal hepatic lobulae (right).

[47, 48] This feature is essential for modelling tissue time-evolution, namely the remodelling and regeneration, or degradation due to evolving pathologies. Coupling processes undergoing at the local (microscopic) scales with physical conservation laws described at the global level is essential and, in general, leads to the structure spatial grading, such as variation of the volume fractions of capillaries, or size of the lobulae. Although, to describe the tissue evolution, linear or linearized micromodels can be used to derive upscaled (homogenized) macro-models, it is important to reflect the dependence of the effective (macroscopic) model parameters on the evolving microconfiguration. For this, the sensitivity analysis approach reported in [49] can well be adapted.

5. **The role of the LRPC in the multiscale hierarchical modelling**. The liver presents a complicated hierarchically organized structure which requires quite different models according to the size of the heterogeneity distinguished at different scales. The LRPC based homogenization can provide just a part of a complete model of the whole liver. As pointed out in the Introduction, the upper-level vasculatures (labelled as PV, HV and HA) can be characterized as perfusion trees arranged hierarchically over several scales in the context of available imaging techniques with limited resolution intervals. Therefore adequate constructive techniques are needed to bridge the scales and to provide complete perfusion trees with limited data, see *e.g.* [14–16, 18] and references cited therein. In [17], a combined 3D–1D approach has been proposed, incorporating a so-called multi-compartment Darcy flow model with 1D flow models accounting for the upper hierarchies of the vascular trees. As a remarkable advantage, the model equations of each compartment associated with a specified hierarchy of the perfusion tree attain the form of Eq (10) obtained at the lobular level characterized as a (locally) periodic structure. This coherency of the two models provides a natural and conceptually simple modelling transition from the bifurcating vasculature to the one represented by the LRPC. Alternatively, the branching vascular structures can be connected with a continuum representation of the parenchyma directly [28].

## Concluding remarks and further challenges

We introduced a parameterized geometrical model of the liver parenchyma unit, the LRPC, which comprises two functional liver units—the portal lobule and the acinus. By translations of the LRPC, a periodic lattice can be generated to construct an idealized lobular structure which, however, can be transformed locally according to the macroscopic position within the whole liver. The LRPC based homogenization must be combined with other modelling approaches which are adequate to treat the flow and also the fluid interaction with a homogenized deforming parenchyma at the macroscopic scale. This issue is far beyond the scope of this paper and has not been discussed.

It has been demonstrated, how the LRPC is applied in the perfusion modelling based on the Double-Darcy model of the homogenized periodic structure. Alternative models can be considered, such as [38, 39, 41], which employ a similar homogenization concept, although a convenient model accounting correctly for the fluid-structure interaction and the blood rheology in the transition between the precapillary vessels and the sinusoidal porosity remains an open problem. As an advantage over the phenomenological approaches, such as the model based on the multi-network poroelasticity theory [50, 51], by virtue of the effective model parameters computed using the characteristic responses of the micromodel defined in the LRPC geometry, the homogenization method enables to respect the geometrical features of the parenchyma. With increasing knowledge of the metabolic and pathology mechanisms of the liver tissue, to develop models for prediction of the tissue regeneration, it appears indispensable to couple the processes undergoing at the microstructure level with the global, macroscopic processes reflecting mass transport and other physical conservation laws relevant to the whole organ level. This can be ensured by the homogenization based modelling which builds on the LRPC providing the periodicity concept of the tissue heterogeneity. To approach realistic tissue morphology, aggregates of perturbed, or distorted LRPC can be used, cf. [20]. Besides that, spatially non-uniform distribution of the lobular structure, the macroscopically varying periodicity and spatial grading can be achieved by transforming the LRPC according to macroscopic position in the liver. A similar concept has been proposed to respect large deformation of the perfused tissue leading to variations of the local micro-configurations [47].

## Appendix

### Notation

In what follows, we employ the Euclidean vector space $\mathbb{E} = \mathcal{E}(\boldsymbol{0}, \boldsymbol{i}_1, \boldsymbol{i}_2, \boldsymbol{i}_3)$ defined by the orthonormal vector basis $\{\boldsymbol{i}_k\}$. To set the notation, any vector $\boldsymbol{a} \in \mathbb{E}$ is determined by the triplet ($a^1$, $a^2$, $a^3$), such that $\boldsymbol{a} = \sum_{i=1}^{3} a^i \boldsymbol{i}_i$, where we use the upper and lower indices $^i$ and $_i$ to distinguish different items, or different vectorial objects. The scalar product of any two vectors $\boldsymbol{a}, \boldsymbol{b} \in \mathbb{E}$ is denoted by $\boldsymbol{a} \cdot \boldsymbol{b}$, while their vector product is $\boldsymbol{a} \times \boldsymbol{b}$. The Euclidean norm of $\boldsymbol{a}$ is $|\boldsymbol{a}| = \sqrt{\boldsymbol{a} \cdot \boldsymbol{a}}$.

### The primary and the dual lattices

These lattices are established using two vectorial bases, $\{\boldsymbol{e}_1, \boldsymbol{e}_2, \boldsymbol{e}_3\}$ and $\{\boldsymbol{g}_1, \boldsymbol{g}_2, \boldsymbol{g}_3\}$, where $\boldsymbol{e}_k, \boldsymbol{g}_l \in \mathbb{E}$, satisfying $|\boldsymbol{e}_k| = 1$ and $|\boldsymbol{g}_l| = 1$, $k, l = 1, \ldots, 3$. Their components are introduced, as follows, see Fig 3,

$$\boldsymbol{e}_1 \cdot \boldsymbol{e}_2 = cos(\pi/3) = 1/2 , \quad \boldsymbol{e}_3 = c\boldsymbol{e}_1 \times \boldsymbol{e}_2 \text{ with } c = 1/sin(\pi/3) \approx 1.1547 ,$$

$$\boldsymbol{g}_1 = (\boldsymbol{e}_1 + \boldsymbol{e}_2)/(2 \cos \pi/6) , \quad \boldsymbol{g}_2 = -\boldsymbol{e}_2 \times \boldsymbol{e}_3 \text{ and } \boldsymbol{g}_3 = \boldsymbol{e}_3 . \tag{15}$$

The *primary lattice* with its characteristic size $\bar{a} > 0$ is generated by the ground structure nodes $\{\boldsymbol{h}^k\}_{k=1,\ldots,4}$ which determine positions of the central veins,

$$\boldsymbol{h}^1 = \boldsymbol{0} \ , \ \ \boldsymbol{h}^k = \bar{a}\boldsymbol{e}_k, \ k = 2, 3 \ , \ \ \boldsymbol{h}^4 = \bar{a}(\boldsymbol{e}_1 + \boldsymbol{e}_2). \tag{16}$$

The *dual lattice* is generated by the ground structure nodes $\{\boldsymbol{p}^l\}_{l=1,\ldots,4}$ which determine positions of the vertex veins,

$$\boldsymbol{p}^1 \ \ = \frac{1}{3}\sum_{k=1}^{3}\boldsymbol{h}^k \ , \ \ \boldsymbol{p}^l = \boldsymbol{p}^1 + \bar{b}\boldsymbol{g}_{l-1} \ , \ \ l = 2, 3 \ , \ \ \boldsymbol{p}^4 = \boldsymbol{p}^1 - \bar{b}(\boldsymbol{g}_1 + \boldsymbol{g}_2) \ , \tag{17}$$

where the dual lattice parameter $\bar{b} = \bar{a}\sqrt{3}/3 \approx 0.5774\bar{a}$. The ground structures are depicted in Fig 3.

## Author Contributions

**Conceptualization:** Eduard Rohan, Václav Liška.

**Formal analysis:** Eduard Rohan.

**Funding acquisition:** Václav Liška.

**Methodology:** Eduard Rohan.

**Project administration:** Václav Liška.

**Software:** Jana Camprová Turjanicová.

**Supervision:** Eduard Rohan.

**Visualization:** Jana Camprová Turjanicová.

**Writing – original draft:** Eduard Rohan, Václav Liška.

**Writing – review & editing:** Eduard Rohan, Jana Camprová Turjanicová.

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
