## [Decision Letter · Decision Letter 0]

3 May 2021

PONE-D-21-02788

Geometrical model of lobular structure and its importance for the liver perfusion analysis

PLOS ONE

Dear Dr. Rohan,

Thank you for submitting your manuscript to PLOS ONE. After careful consideration, we feel that it has merit but does not fully meet PLOS ONE’s publication criteria as it currently stands. Therefore, we invite you to submit a revised version of the manuscript that addresses the points raised during the review process.

The reviewer addressed important questions that need to be clarified. They will be helpful to improve the manuscript.

A rebuttal letter that responds to each point raised by the academic editor and reviewer(s). You should upload this letter as a separate file labeled 'Response to Reviewer'.A marked-up copy of your manuscript that highlights changes made to the original version. You should upload this as a separate file labeled 'Revised Manuscript with Track Changes'.An unmarked version of your revised paper without tracked changes. You should upload this as a separate file labeled 'Manuscript'.

We look forward to receiving your revised manuscript.

Kind regards,

Adélia Sequeira, Ph.D

Academic Editor

PLOS ONE

Journal Requirements:

We note you have included a table to which you do not refer in the text of your manuscript. Please ensure that you refer to Table 1 and 2 in your text; if accepted, production will need this reference to link the reader to the Table.

Reviewers' comments:

Reviewer's Responses to Questions

**Comments to the Author**

1. Is the manuscript technically sound, and do the data support the conclusions?

Reviewer #1: Partly

2. Has the statistical analysis been performed appropriately and rigorously? 

Reviewer #1: N/A

3. Have the authors made all data underlying the findings in their manuscript fully available?

Reviewer #1: Yes

4. Is the manuscript presented in an intelligible fashion and written in standard English?

Reviewer #1: No

5. Review Comments to the Author

Reviewer #1: This manuscript is concerned with the modelling of blood circulation

in the liver tissue. The liver has fundamental units (the lobules),

the structure of which has been considered in various models of liver

blood perfusion. In the present manuscript the authors propose the use

of a smaller unit, which they call representative periodic cell

(LRPC). They discuss how this enables to use homogenisation methods

to derive macroscopic liver perfusion properties.

The paper is quite well written but, in my view, many parts are

unnecessarily over complicated. This makes it reading the manuscript

more difficult that it ought to be and, ultimately, makes it hard to

grasp some of the key features of the work. I would recommend that the

authors simplify the exposition, which does not necessarily imply that

the presentation needs to be less formal and rigorous.

Having said this, I have some fundamental doubts about the work that I

explain in the following.

- To my knowledge most of the resistance to the flow occurs at the

level of the microcirculation, which means in the sinusoids. The

authors' approach allows one to compute the permeability of small

portal and hepatic veins, but obviously not of the sinusoids. If it

is true that flow resistance is concentrated in the sinusoids, this

spoils the usefulness of determining the permeability of larger

vessels.

- I also ave some doubts about the fact that the authors' approach

allows one to model macroscopic variations of the perfusion of liver

tissue. Blood is delivered to microscopic structures through large

vessels, within which the resistance is relatively small. Thus the

view of a macroscopic percolation through a porous medium

constituted by a large number of LRPCs is not very realistic, in my

opinion. Blood is transported to "groups" of LRPCs through large

vessels and only locally at the microscale blood flow can be thought

of as the flow in a porous medium. In order to understand possible

perfusion depletion of certain tissue regions one has to study blood

flow in relatively large vessels.

I think the authors should carefully consider the above points before

the manuscript can be considered for publication on PLOS ONE.

6. PLOS authors have the option to publish the peer review history of their article (what does this mean?). If published, this will include your full peer review and any attached files.

Reviewer #1: No

---

## [Author Response · Author response to Decision Letter 0]

3 Jul 2021

Please, see the pages 2-4 of the Cover letter (PDF file)

OR

the attached PDF entitled: Reply-to-Referee-PLOSone2021-R.pdf

---

## [Decision Letter · Decision Letter 1]

3 Nov 2021

Geometrical model of lobular structure and its importance for the liver perfusion analysis

PONE-D-21-02788R1

Dear Dr. Rohan

We’re pleased to inform you that your manuscript has been judged scientifically suitable for publication and will be formally accepted for publication once it meets all outstanding technical requirements.

Kind regards,

Adélia Sequeira, Ph.D

Academic Editor

PLOS ONE

Additional Editor Comments (optional):

The new version of the manuscript has been significantly improved and I suggest its acceptance in the present form.

We deeply apologize for the delay that certainly was partially due to the COVID-19 pandemic.

Reviewers' comments:

Reviewer's Responses to Questions

**Comments to the Author**

1. If the authors have adequately addressed your comments raised in a previous round of review and you feel that this manuscript is now acceptable for publication, you may indicate that here to bypass the “Comments to the Author” section, enter your conflict of interest statement in the “Confidential to Editor” section, and submit your "Accept" recommendation.

Reviewer #1: All comments have been addressed

2. Is the manuscript technically sound, and do the data support the conclusions?

Reviewer #1: Yes

3. Has the statistical analysis been performed appropriately and rigorously? 

Reviewer #1: N/A

4. Have the authors made all data underlying the findings in their manuscript fully available?

Reviewer #1: Yes

5. Is the manuscript presented in an intelligible fashion and written in standard English?

Reviewer #1: Yes

6. Review Comments to the Author

Reviewer #1: I carefully read the revised version of the manuscript, perusing in

particular the changes with respect to the original submission. The

authors have replied competently and accurately to my comments and

have changed the manuscript accordingly. In my opinion the manuscript

has improved with respect to the original version; this being

particularly true for the Discussion section.

Overall, I think that the paper is carefully written and I suggest

that it is published in the present form.

7. PLOS authors have the option to publish the peer review history of their article (what does this mean?). If published, this will include your full peer review and any attached files.

Reviewer #1: No

---

## [Editor Report · Acceptance letter]

23 Nov 2021

PONE-D-21-02788R1 

Geometrical model of lobular structure and its importance for the liver perfusion analysis 

Dear Dr. Rohan:

I'm pleased to inform you that your manuscript has been deemed suitable for publication in PLOS ONE. Congratulations! Your manuscript is now with our production department. 

Kind regards, 

on behalf of

Dr. Adélia Sequeira 

Academic Editor

PLOS ONE